# A Data-Driven Simulation of the Trophallactic Network and Intranidal Food Flow Dissemination in Ants

**DOI:** 10.3390/ani12212963

**Published:** 2022-10-28

**Authors:** Olivier Bles, Jean-Louis Deneubourg, Cédric Sueur, Stamatios C. Nicolis

**Affiliations:** 1Center for Nonlinear Phenomena and Complex Systems (Cenoli)—CP 231, Université Libre de Bruxelles (ULB), B-1050 Bruxelles, Belgium; 2Université de Strasbourg, CNRS (Centre National de la Recherche Scientifique), IPHC (Institut Pluridisciplinaire Hubert Curien), UMR 7178, 67000 Strasbourg, France; 3Institut Universitaire de France, 75005 Paris, France

**Keywords:** food exchange, interindividual difference, interacting agents, distribution, starvation, *Lasius niger*

## Abstract

**Simple Summary:**

Social insects are able, through decentralised processes, to make collective decisions in various situations, such as decisions regarding the nest choice or exploitation of food sources. At the intranidal level, a network of trophallaxis leads to the dissemination of food, but there are few empirical and far less theoretical studies investigating the mechanisms and the behavioural rules governing the dynamics and the patterns of food exchange at the individual and colony levels in ant colonies. In our study, we developed and analysed a data-driven model whose results fit the characteristics of the observed networks of trophallactic exchange in colonies of Lasius niger ants (a model species). We tested different assumptions concerning the trophallactic behavioural rules at the individual level. The model outcomes and their comparisons with our experiments offer new insights into the link between these individual behaviours and the structure and efficiency of the resulting network of food exchange at the colony level, as well as on the dynamics of the food flow entering the nest. Through a multidisciplinary approach involving theory and experimentation, we validated the model and, in this paper, discuss the biological relevance of our assumptions.

**Abstract:**

Food sharing can occur in both social and non-social species, but it is crucial in eusocial species, in which only some group members collect food. This food collection and the intranidal (i.e., inside the nest) food distribution through trophallactic (i.e., mouth-to-mouth) exchanges are fundamental in eusocial insects. However, the behavioural rules underlying the regulation and the dynamics of food intake and the resulting networks of exchange are poorly understood. In this study, we provide new insights into the behavioural rules underlying the structure of trophallactic networks and food dissemination dynamics within the colony. We build a simple data-driven model that implements interindividual variability and the division of labour to investigate the processes of food accumulation/dissemination inside the nest, both at the individual and collective levels. We also test the alternative hypotheses (no variability and no division of labour). The division of labour, combined with inter-individual variability, leads to predictions of the food dynamics and exchange networks that run, contrary to the other models. Our results suggest a link between the interindividual heterogeneity of the trophallactic behaviours, the food flow dynamics and the network of trophallactic events. Our results show that a slight level of heterogeneity in the number of trophallactic events is enough to generate the properties of the experimental networks and seems to be crucial for the creation of efficient trophallactic networks. Despite the relative simplicity of the model rules, efficient trophallactic networks may emerge as the networks observed in ants, leading to a better understanding of the evolution of self-organisation in such societies.

## 1. Introduction

Food sharing is not the most obvious advantage of group living but can occur in different species [1,2,3]. The evolutionary origins of food sharing have been studied using the predictions of reciprocal altruism [4], biological markets [5] and multilevel selection [6]. Food exchange is central to eusocial species such as ants, termites, bees and even naked mole rats [7,8,9]. In eusocial species, only a restricted number of individuals forage and retrieve food for the rest of the colony, forming trophallaxis (i.e., mouth-to-mouth food exchange) networks [10]. Indeed, insect societies may be seen as social networks whose structure is shaped by individual (nodes) behaviours and interactions (edges) between individuals, including food sharing. While different networks (colonies) may allocate the same total amount of time to a task (such as brood care or food collection), the time investment or the efficiency (i.e., the quantity of food exchanged per interaction) can be relatively different between nodes [11,12] according to their interactions and the emergent hierarchical and modular property of the trophallaxis networks [10]. The topology of social networks drives information transmission [13,14] and food stock building [15] and influences a range of collective outcomes, such as the transmission of parasites and pathogens [16,17,18,19]. Food stock building in social insects involves networks of food-sharing interactions through trophallactic events, during which not only food is transferred [20]. A small fraction of the workers (the foragers) collect the food that is distributed to the nonforaging part of the colony, which in turn disseminates the food [21,22]. A chain of demands, starting with the larvae and the queen, sharpens and adjusts the foraging activity to the colony’s needs [23,24]. No single worker has a comprehensive understanding of the nutritional status of the whole colony. Instead, colony-level nutritional regulation is an emergent property resulting from numerous individual behaviours (e.g., foraging and disseminating) modulated by local information (such as the individual crop content [25]). In such a process, the interindividual variability in the responses (i.e., individuals do not have the same response threshold) may affect the collective outcomes and performance of the colony [26,27,28,29].

Many studies have focused on interindividual variability in searching behaviour and in food collection efforts (e.g., in honeybees [11,30], bumblebees [25] and ants [26,31,32,33,34]). However, the intertwining of the interindividual heterogeneity with the food dissemination activity and the division of work and the resulting trophallaxis networks at the intranidal level remain far less extensively studied. Some empirical works revealed the colony-level dynamics of food sharing and accumulation, as well as the negative feedback that regulates the food flow entering the nest [35,36,37]. The absence of individual identification (i.e., each individual is identified and followed for behavioural scoring) in such studies limits the inquiries regarding the relationship between the level of the workers’ contributions and colony food management. The recent technological improvements in the automation of individual identification [35,37,38,39,40,41] allow for better investigations of the individual behaviours involved in food exchanges. An early study analysing the whole trophallaxis network showed a spatial re-organisation of worker positions in the presence of starvation, accelerating the food stock recovery [15]. Network resilience (i.e., capacity to maintain the food exchange whilst removing central individuals) and efficiency (i.e., how fast the food is transmitted from the foragers to peripheral individuals) are stable in accordance with the colony size but increase in the presence of broods, presumably in response to the nutritional needs of larvae [10]. Other studies focused on the individual behavioural rules regulating the food exchanges [37,40,42,43], refuting some classical assumptions about this phenomenon, which was commonly viewed as a deterministic process [15,35,35,44,45]. In particular, the authors showed that (1) the donor does not deliver its entire crop load, nor does the recipient fill up to its crop capacity; (2) the food flow during a trophallactic event can be bidirectional; and (3) foragers are able to leave the nest even if their crops are not completely empty.

Moreover, a high level of variability is observed in the amount of food transferred during a trophallactic exchange for the crop content of a given recipient as well as in the crop load of the foragers exiting the nest [40]. These variabilities prevent us from drawing any clear conclusions about the relationships between the crop content and the nest-leaving behaviour or the amount of food transferred during an exchange. The consequences of this stochasticity on the food flow dynamics and food spreading speed are not straightforward in terms of their empirical assessment and are therefore overlooked [46].

In the context of foraging activity, a widely accepted behavioural categorisation distinguishes the individuals visiting the food source (even once) and bringing the food back to the nest (foragers) and the individuals staying inside the nest (non-foragers). However, the link between this categorisation and the respective contributions of each caste (foragers vs. non-foragers) in intranidal food dissemination and the characteristics of the trophallactic network are far from clear. The ways in which intra-caste variability in food exchange (i.e., differences in food exchange between foragers and non-foragers) behaviour affects food dissemination within the context of the trophallactic network are also poorly understood.

In this study, we attempt to fill this gap by developing a data-driven model of trophallactic networks that implements interindividual variability and the division of labour. We also tested the alternative hypotheses (no variability and no division of labour). These models are based on empirical data collected on the food exchange process in colonies of *Lasius niger* ants (see Section 3). Our goal was to identify the minimal set of rules governing the trophallactic and foraging behaviours and to capture the main features of the food exchange process in our experimental ant colonies. The model includes four activities: departure from the nest, food collection, travel back to the nest and food exchange (donation/reception).

While some levels of variability are often expected to enhance foraging success [31], here, we explore the effects of two levels of variability in the trophallactic behaviour on food spreading:

(i) The effect resulting from the existence of forager- and non-forager-specific behaviours. We first assume that non-forager and forager trophallactic behaviours are identical and thus consider one behavioural caste (OC for the one-caste assumption), which corresponds to a null hypothesis. Next, we explore the case where a differentiation of non-forager and forager trophallactic behaviours occurs after the first visit of the foragers to the food source and thus consider two behavioural castes (TEC for the two-emergent-caste assumption), which corresponds to the scenario that can be observed in ants.

(ii) The effect occurring within castes. In this case, individuals of each caste have a probability of performing a trophallactic event resulting from a particular distribution. Three common probability distributions are tested, namely, the delta distribution, uniform distribution and exponential distribution, resulting in the probability of variation in the giving or receiving food (explanations are given in Section 2), with exponential distribution being closer to the behaviour observed in forager ants, which give more than they receive, and vice versa with respect to non-foragers.

## 2. Materials and Methods

### Model Description

To capture the essence of the food collection and storage dynamics as well as the intranidal trophallactic network properties of *L. niger**,* we developed an agent-based model. At the beginning of each simulation (*t* = 0), the colony only contains non-forager individual *NFs* with a crop content equal to zero (*Q*_i_(0) = 0). The food source access is unrestricted, and the quantity of food is unlimited. At each timestep *t*, every non-forager in the nest, selected in a random order, can leave the nest and start to feed at the food source with a probability per time unit *α*, which is the inverse of the mean time spent before leaving the nest. Each individual visiting the food source at least once is considered to be a forager *F* for the rest of the simulations. After a certain amount of time spent at the food source, the foragers *Fs*, containing an amount of food *Q* that is proportional to their time spent feeding, return to the nest, with a probability *β* corresponding to the inverse of the time spent feeding. At each timestep *t*, every individual *i* (*i* = *F*, *NF*) in the nest can randomly exchange food, respectively, as a donor or receiver, with a probability *θ_i_* (as donor) or ϒ*_i_* (as receiver), which are increasing (or, respectively, decreasing) functions that depend in a sigmoidal manner on the current crop content $Q_{i}$ [40,42]:(1)$$\theta_{i}=\frac{\theta_{i}\left(0\right)\;\cdot \;Q_{i}^{n}}{k^{n}+Q_{i}^{n}}$$
(2)$${ϒ}_{i}=\frac{{ϒ}_{i}\left(0\right)\;\cdot \;k^{n}}{k^{n}+Q_{i}^{n}}$$
where *k* is a threshold value of the amount of food carried (as individuals may have different thresholds, [40]) and *n* controls the steepness of the functions [38,39]. A high value of *n* leads to a rapidly increasing (or decreasing, respectively, for Equation (2)) probability of giving (or receiving, respectively, for Equation (2)) food when the value of *Q_i_*, the amount of food carried by the individual *i*, approaches the value of *k*. The probability that two individuals will exchange food depends on the product of their individual interaction probabilities. Furthermore, the food flow is directional from the donor to the receiver and cannot be reversed during a single trophallactic event. The probability *Φ* of a trophallactic pair being separated is equal and constant for both individuals. However, when the donor is empty, the trophallactic event stops. For the sake of simplicity, we determined that the quantity of food exchanged is proportional to the duration of the trophallactic event [40]. The individual maximal carrying capacity is not imposed but results from the product of the individual probability *θ* or *ϒ* of giving or receiving food and the probability *Φ* of the separation of a trophallactic pair. The probability of leaving the nest also decreases with the crop content *Q*:(3)$$\alpha_{i}=\frac{\alpha_{i}\left(0\right)\;\cdot \;k^{n}}{k^{n}+Q_{i}^{n}}$$
with *α_i_*(0) representing the maximum probability of leaving (when *Q* = 0). After the first visit to the food source, this maximum probability *α’_i_*(0) increases (*α’_i_*(0) *> α_i_*(0)). The literature suggests only a weak link between the crop content and the probability of food exchange or the probability of leaving the nest (a wide range of the probability of exchanging food or leaving the nest is observed for a given crop content [37,40]). Therefore, we fixed the values of *n* and *k* as follows: *n* = 2 (smooth increase/decrease in the probability of giving/receiving with an increasing crop content) and *k* = 120 (which corresponds to the mean quantity of food exchanged during one trophallactic event—see the next section for details on the experimental procedure). Given these parameter values, individuals leave the nest with a variable crop content, according to the experiments described in [37]. In a simulation, an individual can give (or receive) food several times, until its crop content is full or empty.

We compared the dynamics of food dissemination at the intranidal level and the properties of trophallactic networks in two versions of the model, which differed only in their assumptions concerning the individual probability of giving or receiving food through a trophallactic event between foragers and non-foragers:

A. In the first version, we made a one-caste assumption (OC version). All individuals of the colony were indistinguishable in terms of their probabilities of giving and receiving food through a trophallactic event (*θ_F_* = *θ_NF_* and ϒ*_F_* = ϒ*_NF_*, respectively).

B. In the second version of the model, we made a two-emergent-caste assumption (TEC version). At the beginning of the simulation, all the individuals are non-forager *NFs*. When a non-forager *NF* leaves the nest to go to the food source, it becomes a forager *F* and, having new information about the food availability, its behaviourally then differs from the non-forager *NFs* in terms of the probabilities of giving and receiving food through a trophallactic event (*θ_F_* > *θ_NF_* and ϒ*_F_* < ϒ*_NF_*, respectively).

In both versions of the model, we tested three hypotheses entailing an increasing level of interindividual variability in the probability *θ_i_* of giving and ϒ*_i_* receiving food: First (i), we tested a delta probability distribution, where all the individuals have the same intrinsic probability of exchanging food (this corresponds to a simple but null hypothesis). Next (ii), we tested a uniform distribution of the probability *θ_i_* of the giving and ϒ*_i_* receiving of food between the individuals, with a standard deviation, respectively, equal to the mean probabilities of *θ_i_* and ϒ*_i_*. Finally (iii), we tested the effect of individual variation in the trophallaxis probability on the food flow entering the nest and the individual inequality in terms of the trophallactic activity, since ants vary in their probability of giving/receiving food following a decreasing exponential law ($f\left(x\right)=\varepsilon \;\cdot \;e^{\left(-\frac{x}{\varepsilon }\right)}$, with ε equal to *θ* or ϒ) (this corresponds to what is observed in ants with foragers giving more and receiving less than non-foragers). In (i), (ii) and (iii), the intrinsic individual probabilities of giving and receiving food through a trophallactic event are attributed at the beginning of each simulation and do not change over time in the one-caste (OC) version of the model. In the two-caste (TEC) version, the individual threshold is updated according to a forager (*F*) value after the ant visits the food source for the 1st time. These two values, before and after the 1st visit to the food source, are not correlated.

## 3. Summary of the Behavioural Experiments

From five large mother colonies (>1000 ants) of *L. niger* (collected in Brussels, Belgium, in autumn 2016), we created five queenless and broodless sub-colonies of 50 randomly chosen workers. The ants were individually labelled with an ArucoColor tag (https://sites.google.com/site/usetrackerac/, accessed on 9 November 2019), allowing for the automatic identification of the ants. Each tag was stuck to the abdomen, had a side length of 0.8 mm, weighed 0.1 mg (corresponding to less than 5% of the average mass of an adult worker or less than 10% of the amount of food a worker carries [41]) and was printed on waterproof paper at a resolution of 1200 dpi. The tags were hand-cut using a scalpel and a steel ruler as guide. Following a 5 min acclimatisation period, we did not observe that the labelling impeded the ants’ behaviours, movements or interactions. Each sub-colony was introduced in the experimental setup between 15 and 18 days prior to the first experiment. The setup was composed of a one-chamber nest (56 × 41 × 2 mm) covered by a glass window. This duration was long enough to stabilise the task repartition among individuals (i.e., the foragers were marked with a specific colour, and we observed that they were always the same individuals after two weeks [42]). A single access route (4 × 3 × 2 mm) led to the foraging area (61 × 49 × 2 mm) containing a 0.3 M sucrose solution and water ad libitum. The walls of the foraging area were covered in Fluon^®^ to prevent the ants from escaping. The sub-colonies were kept at 22 ± 3 °C and 60 ± 5% relative humidity, with a 12:12 h constant photoperiod. After 4 days of starvation, we introduced 3 mL of a 1 M sucrose solution. The ants were filmed for 90 min, starting 30 min before the food source introduction. Each colony was tested once. The video data were recorded using a Panasonic^®^ Lumix DMC-GH4-R mounted with a 30 mm Olympus^®^ ED lens, capturing 25 frames/s at the definition of 4180 × 2160 pixels. We discriminated the foragers (*Fs*) from the non-foragers (*NFs*). An individual was considered a forager if it spent at least five consecutive seconds feeding at the food source during the experiment. For each minute, we performed a scan sampling [43] of all the trophallactic interactions, identifying the donor, the receiver and the X and Y spatial positions of the trophallactic event (contact point of the mandibles of both ants). A trophallactic event was recorded when the ants engaged in mandible-to-mandible contact for longer than 5 s. The directionality of the food flow and the roles of the donor and the receiver were determined by the characteristic body posture and the mandible position [40,44]. A trophallactic event involving the same individuals on two or several consecutive scans was considered as a single trophallactic event of a length of two or several minutes. The raw empirical data and codes, as well as the Appendix A, are available from Zenodo (https://doi.org/10.5281/zenodo.6396637, accessed on 9 March 2022). A complete description of the results, as an example of an empirical trophallactic network, is presented in [45].

## 4. Model Calibration and Comparison of the Model Output with the Experimental Results

The model was calibrated using the values of the parameters derived from the experiments (see previous section), with the parameters given in Figure 1 and Table 1, so that the model reproduces the following experimental results: (1) the mean number of foragers; (2) the mean number of trophallactic events; and (3) the proportions of the four different types of trophallactic pairs (F→NF; F→F; NF→NF; NF→F). Therefore, the simulations were run for 3600 timesteps (with each timestep equal to 1 s) using 53 individuals, corresponding to the duration of the experiment and the mean size of the experimental colonies. The simulations were repeated 1000 times for each set of parameters. For each simulation, the start/end time of each trophallactic event, as well as the identity and role of each individual in the trophallactic pairs (the donor/receiver), were extracted. The complete trophallactic network of each simulation (*n* = 1000) and each experiment (*n* = 5) was built, allowing us to analyse and compare the food dissemination dynamics, the properties of the networks of food exchange and the participation of individuals in the trophallactic events. Classical tools of social network analysis in animal societies [47] were also used to characterise the global properties of each trophallactic network, as well as the role of each individual in the network (see the next section for details on the data analysis). The survival curve of the first arrival at the food source in the experiments is well-fitted by the power law distribution of *α_i_*(0)*,* the individual probability of leaving the nest in the model (Appendix A). This suggests that a few individuals have a high probability of visiting the food source, and most of them have a low probability.

We evaluated and compared the goodness-of-fit of these three outputs between the experiments and both versions of the model (OC and TEC), each tested with the three distributions (delta, uniform and exponential) of the probability *θ_i_*(0) of giving and ϒ*_i_*(0) receiving food through a trophallactic event. Only the version of the model (OC or TEC) that best met this first “selection filter” was considered for a more detailed analysis. A local sensitivity analysis of the selected parameter values is provided (Appendix A).

## 5. Statistical and Social Network Analysis

A Mann–Whitney U test (MW) was used to compare the theoretical and experimental numbers of each type of trophallactic pair (*F*→*NF*; *F*→*F*; *NF*→*NF*; *NF*→*F*; Figure 2 and Appendix A). A Kolmogorov–Smirnov test (KS) was used to analyse the deviation between the theoretical and the experimental distributions of the number of trophallactic events among the colony members (Figure 3, Appendix A) and the cumulative number of trophallactic events (Figure 4A and Appendix A). To quantify the degree of inequality in the trophallactic activity among the workers, we plotted the cumulative distribution of the total number of trophallactic events performed in each trial in the form of a Lorenz curve [49] (Figure 4B and Appendix A). Such a curve displays the share of trophallactic activity (Y axis) accounted for by x% of the workers (sorted by the number of trophallactic events per individual) in the colony. A perfectly equitable distribution of the foraging activity would correspond to the line Y = X. The Gini coefficient (Figure 4C and Appendix A) is known as the ratio between the area below the experimental Lorenz curve and the triangular area below the perfect equality case Y = X and provides a measure of the degree of inequality in the distribution of the trophallactic activity, ranging from 0 (perfect equality) to 1 (perfect inequality). A social network analysis was performed using both the theoretical and experimental results. The nodes correspond to individuals (an example of a theoretical network is presented in Appendix A, where red = foragers and green = non-foragers, and it can be directly compared with the empirical network in [45]), and the edges represent trophallactic events directed from the donor to the receiver. The networks were weighted (i.e., including the number of interactions between two nodes) and directed (i.e., including the direction of exchange). Social network analyses were performed at both the individual level and the functional category (foragers/non-foragers) level. The length of the edge conveys no information. At the individual level, we calculated the betweenness, the closeness and the clustering coefficients of each individual [50]. The betweenness centrality (Figure 5C and Appendix A) is an estimate of how important an individual ant is to the promotion of connectivity across the entire colony, and this value can be measured by the number of times that an individual acted as a bridge along the shortest path between two other ants [51]. The closeness centrality (Figure 5B and Appendix A) is based on the distance (measured by shortest paths in terms of the number of nodes) from an individual to every other individual in the colony. The more central an ant is, the lower its total distance is from all the other ants [47]. The clustering coefficient (Figure 5D and Appendix A) allows us to determine the transitivity (number of triangles) in the network, such as that of node pairs that possess many more edges between them than with the other ones [52]. To assess the effect of the network structure on the food spreading speed, we measured the efficiency (Figure 5A and Appendix A), defined as the multiplicative inverse of the shortest path distance between all the pairs of nodes [53,54]. The efficiency measures how fast an entity (here, the food) is transmitted to the network. Concerning the food spreading speed, we compared the mean theoretical T_50_ and experimental T_50_ (the time when half the trophallactic events were realised, Appendix A). To statistically quantify whether the experimental values were different from the simulations (concerning the T_50_, the Gini coefficient and the social networks metrics: the betweenness, closeness clustering and efficiency coefficients), we used Z-tests (ZT) to compare the experimental mean (*n* = 5) with the corresponding theoretical mean (200 mean scores from 5 randomly selected simulations among 1000 simulations). A Kruskal–Wallis (KW) analysis revealed that the degree (KW, *H* = 0.70, *p* > 0.95), out-degree (KW, *H* = 1.83, *p* = 0.77), in-degree (KW, *H* = 0.66, *p* > 0.95), betweenness (KW, *H* = 9.10, *p* = 0.06), closeness (KW, *H* = 3.98, *p* = 0.41) and eigenvector (KW, *H* = 0.93, *p* = 0.91) distributions among the colony members were homogeneous between the five experiments. Therefore, we merged and averaged the experimental results for the calibration of the model and for the comparison between the experimental and theoretical results. The level of significance was set at *p* < 0.05. All simulations were conducted using Python 3.6, and all the analyses were performed using NetworkX 2.1, PyGraphviz 1.4, NumPy 1.14, SciPy 1.0.0 and Matplotlib 2.2.2. We only indicate the main results in the text, but all the statistical analyses are available. The codes and Appendix A are available at https://doi.org/10.5281/zenodo.6396637 (accessed on 9 March 2022).

## 6. Results

### 6.1. Evaluation of the Quality of the Calibration of the Models

The values of the probability *θ* of giving and ϒ receiving food through a trophallactic event that best fit the experimental outputs are presented in Table 1. Both model versions (OC and TEC) closely reproduced the empirically measured number of trophallactic events while reflecting the number of foragers, although they were not explicitly imposed at the beginning of the simulation. This result is independent of the distribution (delta, uniform, exponential) of the individual probability of giving/receiving that was implemented (Table 1).

We then determined the extent to which each type of trophallactic pair (*F*→*F*, *F*→*NF*, *NF*→*F*, *NF*→*NF*) contributes to the total number of trophallactic events. The OC version, regardless of the distribution implemented, systematically reproduced the proportion of each type of trophallactic pair (*F*→*F*, *F*→*NF*, *NF*→*F*, *NF*→*NF*) with a lower accuracy than the TEC version (Figure 2 and Appendix A and Appendix A). The OC version, regardless of the distribution, significantly overestimated the number of *NF*→*NF* exchanges and underestimated the number of *F*→*F* exchanges (MW: *p* < 0.05 in each case; see also Appendix A for details on the statistical analysis), while the TEC version, regardless of the distribution, reproduced the number of each type of trophallactic pairs (MW: *p* > 0.2 in each case; see Figure 2 and Appendix A for details on the statistical analysis).

As the OC version failed to reproduce the experimentally observed proportions of each type of trophallactic pair, in the rest of the paper, we will focus on the TEC version. We expected that this OC version (one-caste assumption) would fail to fit with the observed data, as it is the simplest assumption but not the one found in most colonies of ants where the castes of foragers and non-foragers exist.

### 6.2. Individual Contributions to Food Dispersion/Accumulation

After investigating the trophallactic activity at the colony level, we analysed the way in which each ant participated in the trophallactic exchanges. Our main aim was to understand whether interindividual variability in the trophallactic activity was required to finely tune the experimental distribution of the trophallactic activity/contribution to food dissemination. Figure 3A and Appendix A show the distribution of the number of trophallactic events executed by all the ants, both at the theoretical and experimental levels. While a KS test indicated a significant improvement of the fitting, along with an increasing level of interindividual variability implemented in the model (exponential > uniform > delta), only the TEC exponential version was not significantly different from the experiments. Note that the model (with the delta and uniform distributions) always underestimated the proportion of inactive individuals (Appendix A).

We then focused on the distribution of the number of trophallactic events on a finer scale: the numbers of trophallactic events as per the donors and receivers, respectively, use the same formulation for foragers and non foragers (Figure 3B–E and Appendix A). Only the TEC exponential model was not different from the experimental distribution of the trophallactic activity for each category (KS: *p* > 0.21 in each case; see Appendix A for details on statistical analysis).

### 6.3. Food Spreading, Heterogeneity and Social Network Analysis

No difference was found between the dynamics of food accumulation in the simulations and experiments using the three versions of the model (KS: *p* > 0.95 in all cases; see also Figure 4A and Appendix A for details on the statistical analysis). Concerning the speed of the food spreading, a Z-test indicated no significant difference between the experimental and theoretical T_50_ (time required to reach 50% of the total number of trophallactic events) of the TEC model, regardless of the distribution implemented (Appendix A). The next step consisted of testing the ability of our model to reproduce the experimentally observed interindividual heterogeneity in the food spreading activity, with the majority of the trophallactic events performed by a relatively small number of ants. We statistically quantified the heterogeneity in the food spreading activity between the simulations and experiments using the Lorenz curve and Gini coefficient (Figure 4B,C and Appendix A). Only the TEC exponential model produced a heterogeneity of the food spreading activity as high as that observed in the experiments (ZT: *Z* = −0.9; *p* = 0.39).

We then investigated the characteristics of the trophallactic networks to determine whether the structure of the empirical networks facilitated the food spreading compared with the simulations. Again, only the efficiency of the trophallactic networks resulting from the TEC exponential model was not significantly different from the empirically measured efficiency (ZT; *Z* = −0.9, *p* = 0.38; Figure 5A and Appendix A). Concerning the closeness, betweenness and clustering coefficients, no significant differences were observed between the three distributions (delta, uniform, exponential) in the TEC model and the experiments (ZT; *p* > 0.05; Figure 5B–D and Appendix A). Nevertheless, increasing the theoretical interindividual variability (delta > uniform > exponential) led to a greater level of accordance between the theoretical and experimental results for these three coefficients.

## 7. Discussion

We developed and analysed an agent-based model to investigate the mechanisms underlying the intranidal food spreading in ant nests based on trophallactic exchanges between the colony members. The keystone hypothesis was that no division of labour is at work a priori, as far as the trophallactic exchange is concerned. Rather, the trophallactic processes lead to chains of exchanges based on the random encounters of potential partners. We showed, as expected, that a behavioural shift in the probability of giving/receiving food as a forager in the TEC version of the model and a right-skewed (exponential) distribution of the probability of giving/receiving food among all the colony members (Appendix A) are sufficient for reproducing the trophallactic networks.

The Gini coefficient revealed that the empirical interindividual level of heterogeneity in the ants’ participation in the food dissemination activity, both in the foragers and non-foragers, was only generated by the TEC exponential model (Appendix A). Here, few ants were highly engaged in the trophallactic interactions, which is a common property of many observed networks possessing a scale-free property [55,56,57,58,59]. This interindividual variability is often considered important as far as the resilience is concerned [58], even though the removal of the most engaged nodes can severely disrupt the system [60].

For the TEC model, the distribution of the probability of leaving the nest for the first time, which is the only source of variability, introduced a slight level of heterogeneity in the number of trophallactic events that was enough to generate the properties of the experimental networks and seems to be crucial for the creation of efficient trophallactic networks. Indeed, the efficiency of the experimental networks was only generated by the TEC exponential model, as the TEC delta and TEC uniform models displayed a lower efficiency. All versions of the model assumed random encounters between the ants: if the potential donor (receiver) decides to give (receive) food, a trophallactic exchange occurs. This simple hypothesis was sufficient for generating the experimental efficiency of the trophallactic network. One might assume mechanisms of avoidance/attractiveness between partners that have already exchanged food [61,62]. However, our results suggest that no specific trophallactic pairs occur, except for those resulting from the interindividual variability in the probability of participating in a trophallactic exchange. Therefore, the food accumulated by the ants originates from a large number of randomly encountered nestmates that had regurgitated the food they previously received. Indeed, the model and the experiments showed that approximately 40% of the donated food is given by the non-foragers. The efficiency is a measure of the effectiveness of the diffusion of the information/food. This metric assumes a network of individuals with identical needs, in which the efficiency is maximal as soon as all the individuals are connected with one another. This situation may be far from that of real colonies of social insects, in which the needs may be different between individuals [63] and the diffusion of food must satisfy the individual needs. Thus, the effectiveness of the food dissemination in the colony, based on the measure of classical efficiency, suggests a sub-optimal connectivity of the observed network.

Increasing the interindividual variability in the probability of giving/receiving food in a trophallactic event, while keeping the mean probability constant (i.e., increasing the standard deviation of the uniform distribution), leads to a lower number of exchanges/speed of food accumulation in the nest in a given time-window (Appendix A). Our theoretical results are consistent with a previous theoretical work [64] investigating the relationship between the division of labour (trail-laying behaviour) and efficiency of food recruitment and the subsequent role of positive feedback. These results are in accordance with the general consensus regarding the importance of the division of labour in social insects [65,66] and recent experimental results establishing a link between the within-group behavioural variation and task efficiency (e.g., [67]). When variability increases between individuals, this leads to increased variability in the individual social network measures (degree, betweenness, clustering) and to generally more modular, centralized or efficient networks [68], even if, in some cases, these empirical or theoretical networks are not different from random ones [45]. Thus, in this study, our theoretical networks closely mimic empirical ones. However, we must keep in mind that our model (and that of [64]) does not take into account the various ecological (e.g., food availability, nest structure) and physiological (e.g., temperature) constraints that are omnipresent in such systems and affect the efficiency of the processes.

Other model limitations may have affected the goodness-of-fit of our results. Most obviously, our model does not capture any effect of the intranidal spatial organisation/occupation [69,70] on the dynamics of food collection and dissemination in the colony, which are known to be linked to tasks [71] and to affect the collective response [56]. A recent stochastic spatial model that neglects interindividual differences but shares some features of our hypotheses provides useful insights into the role of space during food dissemination [72]. Note that although the spatial segregation of specialised individuals is believed to optimise the performance of social insects [73,74], our model is still consistent with the experiments.

The presence of heterogeneity in the food dissemination effort in a more complex social context, including a queen and larvae that increase the gradient of the division of work and the heterogeneity in the nutritional needs among the colony members [63,75], still requires further investigation. Among these future experimental and theoretical investigations, priority should be given to the ways in which the colony size affects the global dynamics of food exchange and the resulting trophallactic network. Furthermore, the phenomenological character of our model prevents us from drawing any conclusions about the origin of the observed behavioural variability. Is it an outcome or an underlying driver of behavioural/network interactions? These challenging questions require further theoretical and experimental investigations and are of major interest for researchers aiming to clarify the link between genetics [76], physiology (e.g., proteoms, [42], individual experience [77] and social structure.

To conclude, the agreement between the theoretical and empirical data validates the right-skewed behavioural rules used in the model. This analysis succeeds in accounting for the characteristics of the empirically observed trophallactic networks, without evoking behavioural rules other than the right-skewed distribution of the food dissemination effort, modulated by the individual crop load. These two hypotheses are in agreement with some recent empirical work [37]. Hence, the observed networks of trophallactic events of ant colonies do not seem to rely on complex behavioural rules involving the transfer of various types of information during the food exchange or the ability to count the number of trophallactic events executed. The right-skewed distribution of the food dissemination effort and individual crop load are parameters characterizing the decentralised control and organizational resilience of ants [75]. Such decentralized but hierarchical networks also exist in mammals [76]. Low-clustered but high-robust social networks have also been observed in bees [77] and are due to adaptive age polyethism [78], as can be observed in ants. Such simple, non-linear rules leading to efficient networks in nature have been described, ranging from protein complexes [79] to neural networks [80,81] and organization in social insects [82,83]. These properties increase the group performance, as mentioned by Sueur et al. [84] and described by Fontanari and Rodrigues [85]. The collective cognition behind such complex systems suggests that the topology of trophallactic networks and, more generally, of social networks is selected through individual self-organised rules aiming to optimize the problem-solving competence at the group level, which can be described as “collective social niche construction” [86].

## Figures and Tables

**Figure 1 animals-12-02963-f001:**
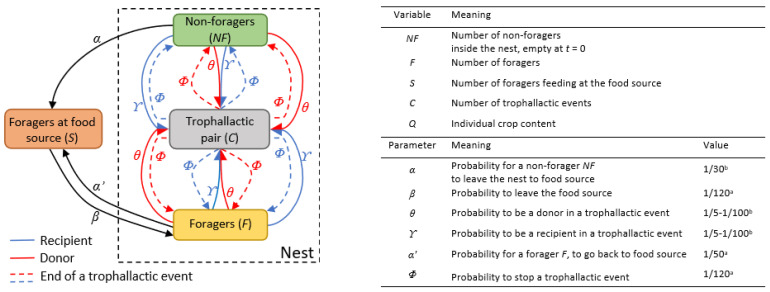
Flow diagram of the model, variables and parameters. The square boxes representing the states of the individuals (*NF*, *F*, *S*, *C*). Black arrows are state transition rates; coloured arrows represent the formation of a trophallactic pair, the red one being the donor and the blue one the receiver. On the right part of the figure, the variables and parameters are defined, and the values of parameters implemented are indicated. ^a^ Parameters estimated from the experiments presented in the Material and Methods section or derived from the literature [46,48]. ^b^ Parameters estimated by fitting in the experiments.

**Figure 2 animals-12-02963-f002:**
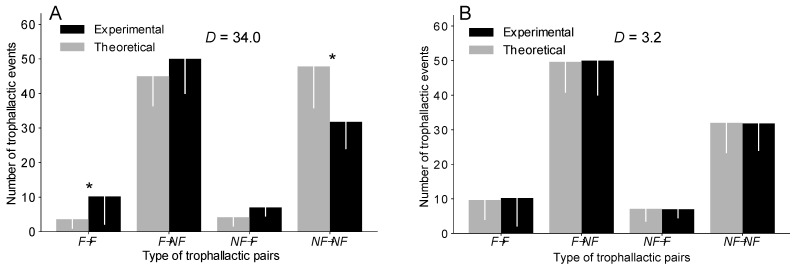
Comparison of the number of each type of trophallactic pairs between the experiments and the simulations implemented based on a delta distribution in the OC version (**A**) and in the TEC version (**B**). * = MW with a *p* < 0.05. Error bar = standard deviation. *D* = sum of the difference in the number of trophallactic pairs of each type (*F*→*F*, *F*→*NF*, *NF*→*F*, *NF*→*NF*) between the experiments and the simulations. See Appendix A for the uniform and exponential distributions and Appendix A for details.

**Figure 3 animals-12-02963-f003:**
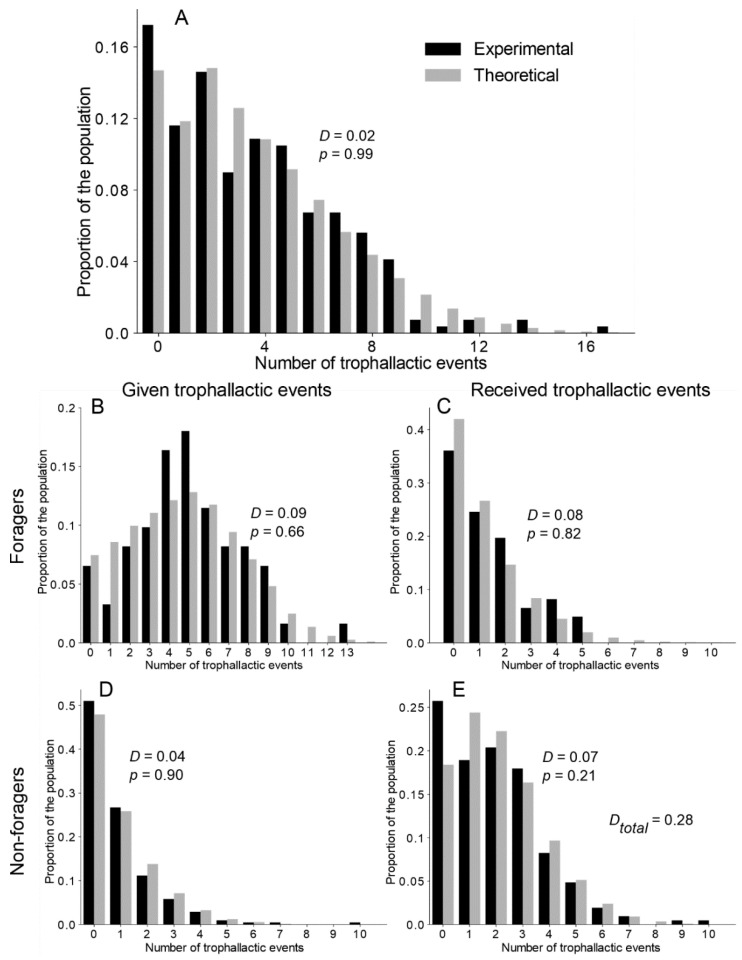
Theoretical and experimental distribution of the number of trophallactic events. (**A**) Distribution of all the trophallactic events at the colony level. (**B**,**C**) Respectively, the given and received trophallactic events performed by the foragers. (**D**,**E**) Respectively, the given and received trophallactic events performed by the non-foragers. Theoretical = TEC exponential model. *D* and *p* on figures = statistical values from the KS test. *D_Total_* = sum of the KS distance (**D**) from the comparison of the distribution in (**B**–**E**). See Appendix A and Appendix A for details on the statistical analysis of the other versions of the model.

**Figure 4 animals-12-02963-f004:**
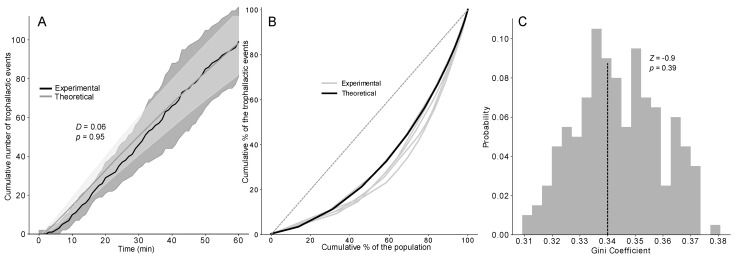
(**A**) Cumulative number of trophallactic events after 1 h of experiments or 3600 timesteps based on 1000 simulations of the TEC exponential model and the experiments (*n* = 5) compared with a KS (*D* and *p*-value). Dashed lines represent the time when 50% of the trophallactic events were realised, both in the model and the experiments. Shaded area = standard deviation. (**B**) Lorenz curves showing the cumulative proportions of trophallactic events (y axis) vs. the individual rank (x axis, sorted by the number of trophallactic events performed by each individual) based on the simulations (black curve, *n* = 1000) and the experiments (grey curves, *n* = 5). (**C**) Distribution of the Gini coefficients from the simulations (grey bars, *n* = 1000) and means from the experiments (dashed lines, *n* = 5), compared with a Z-test (*Z*-value and *p*-value).

**Figure 5 animals-12-02963-f005:**
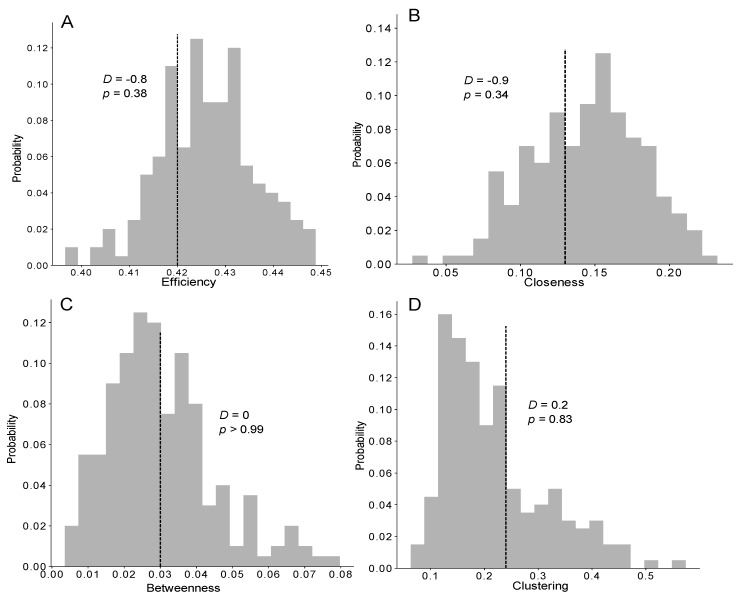
Distribution of the efficiency (**A**), closeness (**B**), betweenness (**C**) and clustering (**D**) coefficients measured in the networks of the TEC exponential model (grey bars, *n* = 1000) and the mean measured from the experiments (vertical dashed line, *n* = 5), compared with a Z-test (*Z*-value and *p*-value). See also Appendix A for the distributions of the network measurements for the other models.

**Table 1 animals-12-02963-t001:** The upper part of the table shows the parameter values of the two versions of the model and three distributions. Only the parameters that vary between models are shown. The lower part of the table shows the main experimental and theoretical results of the foraging activity. See the text for further explanation.

	Parameters	One Caste (OC)			Two Emergent Castes (TEC)		
		Equal	Uniform	Exp.	Equal	Uniform	Exp.
	*θ_F_*	1/11	1/10	1/6	1/10	1/11	1/9
	*θ_W_*				1/33	1/32	1/23
	ϒ*_F_*	1/60	1/50	1/56	1/11	1/9	1/9
	ϒ*_W_*				1/38	1/28	1/27
Results	Experiments (mean ± s.d)						
Number of trophallactic events	99.0 ± 17.4	100.1	98.7	100.5	101.0	99.5	98.7
Number of foragers	12.2 ± 1.9	12.2	12.6	12.2	12.3	12.4	12.5

## Data Availability

Data, codes and Appendix A are available from Zenodo (https://doi.org/10.5281/zenodo.6396637, accessed on 9 March 2022).

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
