# Peer review of "A Data-Driven Simulation of the Trophallactic Network and Intranidal Food Flow Dissemination in Ants"

_animals, 2022, doi:10.3390/ani12212963_

Round 1
Reviewer 1 Report
The manuscript of Bles et al. concerns the link between interindividual heterogeneity of the tropholaxis behavior and the dynamics of food distribution. The authors proposed the mathematical model which was compared with behavioral experiments performed in the laboratory on common garden ant Lasius niger.
The presented study is extended, the paper is very well written. I’m sure the results would be interesting for all myrmecologists dealing with behavior. I have several comments of minor/moderate nature. I have to emphasize that having only basic experience in mathematical modelling I can not fully evaluate the mathematical correctness of the model. My review concerns general soundness of the manuscript and its potential contribution to the existing knowledge on heterogeneity and behavior in ants.
My comments follow
Line 55-57
The sentence is not clear, something is missing
Line 61, 72, 104
Please, specify what do you mean by “the interindividual variability of responses”, “individual identification’, intra-cast variability in food exchange”. For the reader this might be enigmatic.
Line 209
Did ꝊF and ꝨF only differ from ꝊNF and ꝨNF? It seems to me that it should be rather like this:
ꝊF> ꝊNF and
ꝨF< ꝨNF
Line 309 -312
Why you used non-parametric tests instead of parametric ones?
Line 425
Did you do the same for OC version? Figure 4 shows only TEC version.
Line 527
Would be good to propose some examples for “ecological and physiological constraints”
Abstract
Clear “take home message” is lacking. It seems to me that the most important and clear information is in line 493-495. Maybe it is worth to include it somehow in the abstract?
Author Response
Reviewer 1
Comment: The manuscript of Bles et al. concerns the link between interindividual heterogeneity of the tropholaxis behavior and the dynamics of food distribution. The authors proposed the mathematical model which was compared with behavioral experiments performed in the laboratory on common garden ant Lasius niger. The presented study is extended, the paper is very well written. I’m sure the results would be interesting for all myrmecologists dealing with behavior. I have several comments of minor/moderate nature. I have to emphasize that having only basic experience in mathematical modelling I can not fully evaluate the mathematical correctness of the model. My review concerns general soundness of the manuscript and its potential contribution to the existing knowledge on heterogeneity and behavior in ants.
Answer: Thanks you very much for your nice comments. We answered all of the positively.
Comment: Line 55-57: The sentence is not clear, something is missing
Answer: we changed the sentence to “A chain of demands, starting from larvae and the queen, sharpens and adjusts the foraging activity to the colony’s needs” lines 70-71
Comment: Line 61, 72, 104: Please, specify what do you mean by “the interindividual variability of responses”, “individual identification’, intra-cast variability in food exchange”. For the reader this might be enigmatic.
Answer: Done
- interindividual variability of the responses (i.e., individuals do not have the same threshold of response) Lines 76-77
- individual identification (i.e., each individual are identified and followed for behav-ioural scoring), lines 84-85
- intra-caste variability in food exchange (i.e., differences in food-exchange between foragers and non-foragers) lines 114-115
Comment: Line 209: Did ꝊF and ꝨF only differ from ꝊNF and ꝨNF? It seems to me that it should be rather like this:
ꝊF> ꝊNF and
ꝨF< ꝨNF
Answer: Done, line 202
Comment: Line 309 -312: Why you used non-parametric tests instead of parametric ones?
Answer: Because variables are counting and limited in their distributions, so not following Gaussian laws.
Comment: Line 425: Did you do the same for OC version? Figure 4 shows only TEC version.
Answer: Yes, but in the main text, we only presented main and significant results. Remaining results are in the supplementary information. We indicated it lines 373-374.
Comment: line 527: Would be good to propose some examples for “ecological and physiological constraints”
Answer: We added them: ecological (e.g. food availability, nest structure) and physiological (e.g. temperature), lines 524-525
Comment: Abstract: Clear “take home message” is lacking. It seems to me that the most important and clear information is in line 493-495. Maybe it is worth to include it somehow in the abstract?
Answer: Done.
Reviewer 2 Report
The paper in overall very good. The state of the art in complete and focus the interest of the study presented. The objectives are clear, the methodology was adequate, the results are well presented and support the conclusions that cover partially the objectives and knowledge gaps signaled.
Therefor I recommend the publication after the following changes:
The main remark involves the presentation of results, with reference to figure 1 and table 1, within the Materials and Methods chapter (lines 263-264) and statistical results (lines 350-357).
I recommend to authors to describe the assays and the tests in Materials and Methods and present this results in be begging of the Results respective sessions. For instance, present Fig 1 with Table 1, in line 264.
Line 84 - Bonavita-Cougourdan - C in capital
Line 267 - (F→NF; F→ ; NF→ ; NF→F), should be (F→NF; F→ F; NF→ NF; NF→F).
Line 380-381 - The authors should discuss why OC version failed to reproduced experimental observations, either after the sentence or in discussion, if references should be cited to support explanation.
Line 563 - "...the experiments: ..."
I congratulate the authors for the experiment and manuscript quality
Author Response
Reviewer 2
Comment: The paper in overall very good. The state of the art in complete and focus the interest of the study presented. The objectives are clear, the methodology was adequate, the results are well presented and support the conclusions that cover partially the objectives and knowledge gaps signaled.
Answer: Thanks you very much for your nice comments. We answered all of the positively.
Comment: The main remark involves the presentation of results, with reference to figure 1 and table 1, within the Materials and Methods chapter (lines 263-264) and statistical results (lines 350-357). I recommend to authors to describe the assays and the tests in Materials and Methods and present this results in be begging of the Results respective sessions. For instance, present Fig 1 with Table 1, in line 264.
Answer: Done, we moved the figure 1 and the table 1 as you proposed
Comment: Line 84 - Bonavita-Cougourdan - C in capital
Answer: Done
Comment: Line 267 - (F→NF; F→ ; NF→ ; NF→F), should be (F→NF; F→ F; NF→ NF; NF→F).
Answer: Done
Comment: Line 380-381 - The authors should discuss why OC version failed to reproduced experimental observations, either after the sentence or in discussion, if references should be cited to support explanation.
Answer: We added the sentences: “We expected that this OC version (One caste assumption) failed to fit with the observed data as it is the simplest assumption but not the one found in most of ants where castes of foragers and non-foragers exist.” lines 396-398
Line 563 - "...the experiments: ..."
Done
I congratulate the authors for the experiment and manuscript quality
Thank you very much!
Reviewer 3 Report
Overall, I quite liked your paper - which describes an agent-based model (based upon empirically-derived behavior of Lasius niger ants). Below are a few comments and suggestions:
Overall, the paper is good - with a thorough literature review in the intro and discussion, the model is explained pretty well, and the overall premise is good. I think I wasn't too surprised by the relative success of your TEC version (over the OC version) of the model. Division of labor is one of the hallmarks of the success of ant societies. Nor was I too surprised by the success of the right-skewed (exponential) distribution to give/receive food. Again, this seems to be a hallmark of ant societies - where a small number of individuals can be responsible for most instances of a particular sub-task being completed. But the result of your project could also be considered impressive in that with such a small number of behavioral rules and a probabilistic framework in your agent-based model, you successfully recreate many of the properties of real-world ant trophallactic networks. In your discussion, you describe some of the real-world variety of ant societies that your paper wasn't able to address.
A few specific points/ questions:
Line 172 and 173: 'A high value of n leads to a rapidly decreasing (increasing) probability to accept (give) food when the value of Qi, the amount of food carried by the individual i, approaches the value of k.'
I was confused by the 'increasing' and 'give' in parentheses. Can you clarify that?
Line 239: 'This duration was long enough to stabilize the task repartition...' How did you determine this?
In the Model Calibration section, there seems to be a typo or error in Line 267. Shouldn't it read (F arrow NF; F arrow F; NF arrow NF; NF arrow F)?
Line 281: Shouldn't this read, 'This suggests that a few individuals...'
Line 522: 'These results deviate from the general agreement of the importance of division of labor...' But don't your results emphasize the importance of a division of labor in terms of the TEC model being the best fit for your data? As I understand it, your TEC model is a model of division of labor. 'Two emergent castes' are essentially specialized groups of ants, right?
Line 525: I would suggest reconsidering citing any works by J. Pruitt. Although this particular paper may not have been retracted at this date, his pattern of data manipulation/ fraud seems to have extended back to his research in grad school. This is the whole recent Pruittgate imbroglio...
Figure 3c has an error in it. It should say, 'Received'.
Discussion: I do think that there are many more questions one could ask of a model such as this. For example, many trophallaxis interactions (in some species) are not one to one but rather one to multiple (as several ants concurrently take part in trophallaxis with the food donor). How would this affect the results of your model? But you are realistic in how broadly your current model might be applied (while acknowledging the importance of such factors as spatial organization and other real world phenomena that your model does not address).
Author Response
Reviewer 3
Comment: Overall, I quite liked your paper - which describes an agent-based model (based upon empirically-derived behavior of Lasius niger ants). Below are a few comments and suggestions: Overall, the paper is good - with a thorough literature review in the intro and discussion, the model is explained pretty well, and the overall premise is good. I think I wasn't too surprised by the relative success of your TEC version (over the OC version) of the model. Division of labor is one of the hallmarks of the success of ant societies. Nor was I too surprised by the success of the right-skewed (exponential) distribution to give/receive food. Again, this seems to be a hallmark of ant societies - where a small number of individuals can be responsible for most instances of a particular sub-task being completed. But the result of your project could also be considered impressive in that with such a small number of behavioral rules and a probabilistic framework in your agent-based model, you successfully recreate many of the properties of real-world ant trophallactic networks. In your discussion, you describe some of the real-world variety of ant societies that your paper wasn't able to address.
Answer: Thanks you very much for your nice comments. We answered all of the positively. We modified the discussion in order to be clearer
Comment: Line 172 and 173: 'A high value of n leads to a rapidly decreasing (increasing) probability to accept (give) food when the value of Qi, the amount of food carried by the individual i, approaches the value of k.' I was confused by the 'increasing' and 'give' in parentheses. Can you clarify that?
Answer: it depends whether it is from equation 1 and 2. We added the details.
Comment: Line 239: 'This duration was long enough to stabilize the task repartition...' How did you determine this?
Answer: we added this sentence: (i.e., foragers are marked with a specific color and we observed that they are always the same individuals after two weeks [56])
Comment: In the Model Calibration section, there seems to be a typo or error in Line 267. Shouldn't it read (F arrow NF; F arrow F; NF arrow NF; NF arrow F)?
Answer: Corrected
Comment: Line 281: Shouldn't this read, 'This suggests that a few individuals...'
Answer: done.
Comment: Line 522: 'These results deviate from the general agreement of the importance of division of labor...' But don't your results emphasize the importance of a division of labor in terms of the TEC model being the best fit for your data? As I understand it, your TEC model is a model of division of labor. 'Two emergent castes' are essentially specialized groups of ants, right?
Answer: Indeed, we corrected the sentence.
Comment: Line 525: I would suggest reconsidering citing any works by J. Pruitt. Although this particular paper may not have been retracted at this date, his pattern of data manipulation/ fraud seems to have extended back to his research in grad school. This is the whole recent Pruittgate imbroglio...
Answer: Done, we removed the reference.
Comment: Figure 3c has an error in it. It should say, 'Received'.
Answer: Thanks! Corrected.
Comment: Discussion: I do think that there are many more questions one could ask of a model such as this. For example, many trophallaxis interactions (in some species) are not one to one but rather one to multiple (as several ants concurrently take part in trophallaxis with the food donor). How would this affect the results of your model? But you are realistic in how broadly your current model might be applied (while acknowledging the importance of such factors as spatial organization and other real world phenomena that your model does not address).
Answer: Maybe the description of the model was not clear, but in a simulation, an individual can give (or receive) several times food, until its crop content is full or empty. We added this information lines 188-189.
Reviewer 4 Report
I found an easy to follow study in the Introduction, M&M and Results sections, but the Discussion section needs a little additional work.
- please standardize trophallactic pairs notation (L267 vs L311)
- L413: remove duplicate dot;
- first four paragraphs from Discussion section are essentially results; further, the 6th again return to results without compare with published literature;
- Line 525: do you mean (e.g, Pruitt and Riechert 2011; Modlmeier et al. 2012)?
- Line 527: why do you need double parenthesis?
- please, carefully revise the References section because it is very complicated; is it a reference manager problem? Note, for example, lack of italic font in scientific names, abbreviated vs. non-abbreviated journals, capital letters, among others.
- L539: you start paragraph with 'in summary', but there are 24 lines in there, making this paragraph hard to follow and understand which is in summary;
- L 119-124: what OC and TEC stand for? If OC = one behavioural caste and TEC = two behavioural castes, it is not immediately clear what they mean (and we can understand that only in Table 1 - one caste and two emergent castes).
Author Response
Reviewer 4
Comment: I found an easy to follow study in the Introduction, M&M and Results sections, but the Discussion section needs a little additional work.
Answer: Thanks you very much for your nice comments. We answered all of the positively.
Comment: please standardize trophallactic pairs notation (L267 vs L311)
Answer: Done
Comment: L413: remove duplicate dot;
Answer: Done
Comment: first four paragraphs from Discussion section are essentially results; further, the 6th again return to results without compare with published literature;
Answer: We removed these paragraphs from the discussion
Comment: Line 525: do you mean (e.g, Pruitt and Riechert 2011; Modlmeier et al. 2012)?
Answer: corrected
Comment: Line 527: why do you need double parenthesis?
Answer: Corrected.
Comment: please, carefully revise the References section because it is very complicated; is it a reference manager problem? Note, for example, lack of italic font in scientific names, abbreviated vs. non-abbreviated journals, capital letters, among others.
Answer: Done
Comment: L539: you start paragraph with 'in summary', but there are 24 lines in there, making this paragraph hard to follow and understand which is in summary;
Answer: Indeed, this is not a summary but a conclusion, we changed it and we moved the paragraph to the end of the discussion.
Comment: L 119-124: what OC and TEC stand for? If OC = one behavioural caste and TEC = two behavioural castes, it is not immediately clear what they mean (and we can understand that only in Table 1 - one caste and two emergent castes).
Answer: Thanks! We added the definition.
Reviewer 5 Report
The present paper introduces an agent based model for food dissemination dynamics within ant colonies. The model suggests that individual variation in trophallaxis probability and division of labour are sufficient to reproduce the food dynamics and exchange networks seen in experimental populations.
Overall the network analysis and statistical assessments are sound and meaningful for the conclusions drawn.
Comments:
I’ld love to see an example of an empirical network and a simulated network, sometimes visualization helps to illuminate important similarities and differences and offer intuition for why specific measures may differ.
Fig.5 does a great job supporting the TEC Exponential model, but it doesn’t help show how poor the OC/different probability versions perform. On the same subplots, can you also show the distributions of network statistics for one or two of the other variations?
The other important facet is the differences in number of trophallactic events which equates to differing degree distributions for the social networks. Is the degree distribution alone sufficient to produce the measured efficiency, closeness, betweenness, and clustering? Or is this a feature of the interaction probabilities and division of labor from the model? You could test by randomizing the edges using a configuration model and measuring the network statistics.
Author Response
Review 5:
Comment: The present paper introduces an agent based model for food dissemination dynamics within ant colonies. The model suggests that individual variation in trophallaxis probability and division of labour are sufficient to reproduce the food dynamics and exchange networks seen in experimental populations.
Overall the network analysis and statistical assessments are sound and meaningful for the conclusions drawn.
Answer: Thank you very much for your nice comments!
Comment: I’ld love to see an example of an empirical network and a simulated network, sometimes visualization helps to illuminate important similarities and differences and offer intuition for why specific measures may differ.
Answer: Of course, it is already present. An example of a theoretical network is presented in Figure S2 of the supplementary material. Do you have access to it? It can be compared to the empirical network of the study Planckaert et al. 2019 in open access and introduced in the methods section of our paper.
Comment Fig.5 does a great job supporting the TEC Exponential model, but it doesn’t help show how poor the OC/different probability versions perform. On the same subplots, can you also show the distributions of network statistics for one or two of the other variations?
Answer: The figure you ask is very huge and already present in figure S4 of the supplementary material. Did you have access to it? We don’t think that it could fit into the main manuscript. We added this detail in the main manuscript.
Comment: The other important facet is the differences in number of trophallactic events which equates to differing degree distributions for the social networks. Is the degree distribution alone sufficient to produce the measured efficiency, closeness, betweenness, and clustering? Or is this a feature of the interaction probabilities and division of labor from the model? You could test by randomizing the edges using a configuration model and measuring the network statistics.
Answer: Indeed, this question is important and was already tested in other studies and particularly in studies we already did in ants as the ones cited in the manuscript:
Quque, M., Bles, O., Bénard, A., Héraud, A., Meunier, B., Criscuolo, F., ... & Sueur, C. (2021). Hierarchical networks of food exchange in the black garden ant Lasius niger. Insect science, 28(3), 825-838.
Planckaert, J., Nicolis, S. C., Deneubourg, J. L., Sueur, C., & Bles, O. (2019). A spatiotemporal analysis of the food dissemination process and the trophallactic network in the ant Lasius niger. Scientific reports, 9(1), 1-11.
But also many others:
Romano, V., Duboscq, J., Sarabian, C., Thomas, E., Sueur, C., & MacIntosh, A. J. (2016). Modeling infection transmission in primate networks to predict centrality‐based risk. American Journal of Primatology, 78(7), 767-779.
Romano, V., Shen, M., Pansanel, J., MacIntosh, A. J., & Sueur, C. (2018). Social transmission in networks: global efficiency peaks with intermediate levels of modularity. Behavioral ecology and sociobiology, 72(9), 1-10
When variability increases between individuals, this conducts to increased variability in individual social network measures (degree, betweenness, clustering) and to generally more modular, centralized or efficient networks even if sometimes these empirical or theoretical networks are not different from random ones. Degree is generally correlated with all other individual network measures and the degree variability influences the topology of the network (e.g. star network or equal/all connected networks). We added this in the manuscript.
